# *Portulaca oleracea* L. Extract Regulates Hepatic Cholesterol Metabolism via the AMPK/MicroRNA-33/34a Pathway in Rats Fed a High-Cholesterol Diet

**DOI:** 10.3390/nu14163330

**Published:** 2022-08-14

**Authors:** Sojeong Jang, Mak-Soon Lee, Sun-A Kang, Chong-Tai Kim, Yangha Kim

**Affiliations:** 1Department of Nutritional Science and Food Management, Ewha Womans University, Seoul 03760, Korea; 2Graduate Program in System Health Science and Engineering, Department of Nutritional Science and Food Management, Ewha Womans University, Seoul 03760, Korea; 3R&D Center, EastHill Corporation, Suwon 16642, Korea

**Keywords:** *Portulaca oleracea* L., cardiovascular disease, cholesterol metabolism, AMPK, microRNA

## Abstract

This study examined the effect of extruded *Portulaca oleracea* L. extract (PE) in rats fed a high-cholesterol diet through the AMP-activated protein kinase (AMPK) and microRNA (miR)-33/34a pathway. Sprague–Dawley rats were randomized into three groups and fed either a standard diet (SD), a high-cholesterol diet containing 1% cholesterol and 0.5% cholic acid (HC), or an HC diet containing 0.8% PE for 4 weeks. PE supplementation improved serum, liver, and fecal lipid profiles. PE upregulated the expression of genes involved in cholesterol efflux and bile acids’ synthesis such as liver X receptor alpha (LXRα), ATP-binding cassette subfamily G5/G8 (ABCG5/8), and cholesterol 7 alpha-hydroxylase (CYP7A1), and downregulated farnesoid X receptor (FXR) in the liver. In addition, hepatic gene expression levels of apolipoprotein A-l (apoA-1), paraoxonase 1 (PON1), ATP-binding cassette subfamily A1/G1 (ABCA1/G1), lecithin-cholesterol acyltransferase (LCAT), and scavenger receptor class B type 1 (SR-B1), which are related to serum high-density lipoprotein cholesterol metabolism, were upregulated by PE. Furthermore, hepatic AMPK activity in the PE group was higher than in the HC group, and miR-33/34a expression levels were suppressed. These results suggest that PE improves the cholesterol metabolism by modulating AMPK activation and miR-33/34a expression in the liver.

## 1. Introduction

Cardiovascular disease (CVD) remains one of the biggest causes of death worldwide and has emerged as an immense health and economic burden globally [1]. Hypercholesterolemia, characterized by an increase in the serum levels of triglyceride (TG), total cholesterol (TC), and low-density lipoprotein cholesterol (LDL-C), as well as decreased levels of high-density lipoprotein cholesterol (HDL-C), is a well-known risk factor for CVD [2]. Given this close association between circulating lipid levels and CVD, regulating cholesterol metabolism may be a possible target for preventing and treating CVD.

Cholesterol homeostasis is maintained through biliary excretion, which is brought about by converting cholesterol to bile acid (BA) in the liver, which is excreted through the feces [3]. Cholesterol metabolism and BA synthesis are associated with several transcriptional modulators and enzyme activity. Upon excessive intake of dietary cholesterol, liver X receptor α (LXRα), a nuclear hormone receptor, promotes the expression of ATP-binding cassette protein G5/G8 (ABCG5/8) in the liver and intestine [4]. ABCG5/8 serves as a transporter protein for transporting BA from the liver to the intestinal lumen to facilitate fecal excretion [5]. LXRα also regulates cholesterol 7α-hydroxylase (CYP7A1), a rate-limiting enzyme for BA biosynthesis. The farnesoid X receptor (FXR) inhibits CYP7A1 expression as a transcriptional regulator, inhibiting excessive BA synthesis [4]. One important pathway for cholesterol-mediated efflux from macrophage foam cells involves interaction between the ATP-binding cassette transporter A1 (ABCA1) with cholesterol-deficient and cholesterol-depleted apolipoprotein A-1 (apoA-1), forming nascent HDL [6]. Subsequently, the ATP-binding cassette transporter G1 (ABCG1) mediates macrophage cholesterol efflux through interactions with nascent HDL and subsequent cholesterol esterification by lecithin cholesterol acyl transferase (LCAT) to form mature HDL-C [7]. The scavenger receptor class B type 1 (SR-B1) involves selective uptake of cholesteryl esters from mature HDL-C by the liver [8]. In addition, paraoxonase 1 (PON1) is an HDL-associated protein that binds to HDL-C and is mechanistically linked to atherosclerosis [9].

AMP-activated protein kinase (AMPK) is a key lipid and energy metabolism regulator, and activated AMPK promotes vascular health, contributing to the anti-atherosclerotic effect [10,11]. In addition, activation of AMPK in the early stages of atherosclerosis can restore cholesterol homeostasis [12]. Therefore, AMPK is emerging as a promising molecular target for the treatment of CVD.

*Portulaca oleracea* L. (PO) is a herbaceous annual grassy plant with a cosmopolitan distribution belonging to the *Portulacaceae* family [13]. PO has been used as a traditional medicine for centuries in many countries [14]. It has many bioactive components, including polyphenols, flavonoids, alkaloids, vitamins, minerals, and omega-3 fatty acid [13,15,16,17], and it is the most abundant in alpha-linolenic acid among vegetables [17]. In particular, PO has exhibited anti-inflammatory, antioxidant, antitumor, antidiabetic, antiobesity, and hypocholesterolemic effects [13,15,16,18,19]. However, the mechanism of action of PO extract in regulating AMPK and microRNAs (miRs) and its impact on hepatic cholesterol metabolism remain unclear. Therefore, we investigated whether extruded PO extract (PE) affects cholesterol metabolism in the liver of rats fed a high-cholesterol diet. In addition, we evaluated the hepatic AMPK activity and expression of miR-33 and miR-34a.

## 2. Materials and Methods

### 2.1. Preparation of PE

PE was kindly supplied by EastHill (Suwon, Korea). *Portulaca oleracea* L. was ground in a pin mill and passed through an 80-mesh screen. The powder was extruded through a co-rotating and inter-meshing type twin-screw extruder (Buhler Brothers Co., Biex-DNDL 44, Uzwil, Switzerland) with D-44 mm and L/D = 20. The die was a single circular orifice with a 2 mm nozzle, and the barrel temperature was maintained at 160 °C. The feed rate of materials was 6 kg/h, the screw speed was 160 rpm, and the moisture content was 20%. After extrusion, the extrudate was dried at 105 °C for 4 h using a forced-air oven and ground into small particles using a laboratory mill equipped with 0.5 mm mesh screens. Next, 10 kg of ground PO extrudate was placed in an extraction tank with an agitator containing tap water (300 L) and agitated for 4 h at 85 °C. Following this step, the slurry was separated by centrifugation (Supra 30K, Hanil Scientific, Seoul, Korea) at 3500× *g* for 5 min. The supernatant was sterilized at 85 °C for 30 min, and then a powder sample was obtained using a pilot-scale spray dryer (Niro Atomizer, Søborg, Denmark). The powder was stored at 20 °C until use.

### 2.2. Sample Preparation for HPLC Analysis

As phenolic compound standards, gallic acid, chlorogenic acid, ferulic acid, and rutin were purchased from Sigma-Aldrich (St. Louis, MO, USA) and used for the determination of phenolic content in PE. Here, 0.01 g of the standard was precisely weighed out in a 50 mL conical tube, 10 mL of methanol was added to yield a standard concentration of 1000 ppm, and it dissolved thoughly in a vortex mixer for 5 min. It was then diluted to appropriate concentrations ranging from 62.5 to 125, 250, and 500 ppm. Finally, the dilutes were filtered through a 0.20 µm polytetrafluoroethylene syringe filter. In order to determine the phenolic content of PE, 0.1 g of PE powder was accurately weighed out in a 50 mL conical tube, and 10 mL of distilled water was added. It was then dissolved in a vortex mixer for 5 min. Finally, the solution was filtered through a 0.20 µm polytetrafluoethylene syringe filter. Phenolic compounds in samples were quantified based on calibration with standards.

### 2.3. High-Performance Liquid Chromatography (HPLC) Analysis

Phenolic compounds in PE were identified and quantified using the Agilent 1100 HPLC System (Santa Clara, CA, USA). Separation was achieved on a reversed-phase column (Agilent Eclipse XDB-C18; 5 µm, 4.6 × 250 mm) maintained at 30 °C. The injection volume was maintained at 20 µL throughout the experiments. The flow rate of the mobile phase was 1 mL/min, and the analysis was carried out at a wavelength of 280 nm. Gradient elution was employed using two binary systems: A, 2% phosphoric acid in water, and B, acetonitrile. The running time was set at 90 min. The gradient program was as follows: mobile phase held at 0% B for 3 min, gradually 4% B at 15 min, 10% B at 30 min, 15% B at 50 min, 23% B at 60 min, 23% B at 66 min, 25% B, 50% B at 80 min, 80% B at 83 min, and 0% B at 85 min, held up to 90 min. The HPLC chromatogram of the phenolic compounds is presented in Appendix A. The regression equation was obtained from the analysis results of phenolic compound standards (Appendix A). The phenolic content of PE was obtained by substituting y as area and x as quantity. As a result of the HPLC analysis of PE, the peaks of gallic acid at 9.402 min, chlorogenic acid at 31.560 min, ferulic acid at 48.804 min, and rutin at 55.649 min were confirmed.

### 2.4. Animals and Diet

Six-week-old male Sprague–Dawley rats (180–200 g, *n* = 24) were purchased from Doo Yeol Biotech (Seoul, Korea). They were housed at 25 °C with a 12 h light/12 h dark cycle and acclimatized for 1 week with free access to water and a normal chow diet (Harlan 2018S rodent diet, Harlan, Madison, WI, USA). The rats were divided into three groups (*n* = 8 per group) as follows: (1) standard diet (SD), (2) SD supplemented with 1% cholesterol and 0.5% cholic acid (HC), and HC supplemented with 0.8% PE (Table 1). All experimental procedures were approved by the Institutional Animal Care and Use Committee (IACUC) of Ewha Woman’s University (IACUC No. 21-045). At the end of the 4-week PE supplementation, all rats were anesthetized with a mixture of Zoletil 50 (Virbac Laboratories, Carros, France) and Rompun (Bayer Korea, Seoul, Korea) after 12 h overnight fasting. Liver tissue was excised through an abdominal midline incision, weighed immediately, frozen in liquid nitrogen, and stored at −70 °C. Blood samples were collected by cardiac puncture, separated by centrifugation at 3000 rpm for 20 min, and stored at −70 °C.

### 2.5. Determination of Serum Metabolic Parameters

Serum alanine transaminase (ALT) and aspartate transaminase (AST) activities were measured using enzymatic colorimetric commercial kits (Asan Pharmaceutical Co., Seoul, Korea). Serum TC, TG, and HDL-C concentrations were measured using enzymatic colorimetric commercial kits (Embiel, Gunpo, Korea). The LDL-C concentration was calculated by Friedewald’s formula [21] as follows: LDL-C = TC − HDL-C − (TG/5).

### 2.6. Hepatic and Fecal Lipid Extraction

Hepatic and fecal lipids were extracted using the modified Bligh and Dyer method [22]. Homogenates of liver and fecal matter (0.5 g) were placed together with 7.5 mL of chloroform and methanol mixture (1:2, *v*/*v*; Duksan, Ansan, Korea) and vortexed for 20 min. Chloroform (2.5 mL) was added, and the homogenates were centrifuged at 3000 rpm for 20 min. The clear lower phase was collected, dried, weighed, and resuspended in a mixture of n-hexane/isopropanol (3/2, *v*/*v*; Duksan). TG and TC levels in the liver and fecal matter were quantified using enzymatic colorimetric commercial kits (Embiel, Gunpo, Korea).

### 2.7. Fecal BA Analysis

Total bile acid (TBA) content in fecal lipid was measured using a TBA assay kit (BioVision, Milpitas, CA, USA) following an enzymatic cycling method in the presence of nicotinamide adenine dinucleotide hydrogen (NADH). Absorbance was measured at 405 nm with a Varioskan plate reader (Thermo Scientific, Waltham, MA, USA).

### 2.8. Histological Analysis

Histological analysis in the liver tissue was measured as previously described [23]. The fixed liver tissues in 10% formalin solution for 24 h at room temperature were embedded in paraffin blocks, sliced into appropriate portion by a microtome (Leica-microsystems, Wetzlar, Germany), and stained with hematoxylin–eosin (H&E). Digital images of stained liver tissue sections were observed with a microscope (Olympus, Tokyo, Japan) at 400× magnification.

### 2.9. Real-Time Quantitative Polymerase Chain Reaction (RT-qPCR)

Total RNA was isolated from the liver tissues using Ribo Ex total RNA solution (Geneall Biotechnology Co., Daejeon, Korea). Complementary DNA (cDNA) was synthesized using Moloney Murine Leukemia Virus (MMLV) reverse transcriptase kit (Bioneer, Daejeon, Korea). RT-qPCR was performed using a fluorometric thermal cycler (Rotor-Gene Q, Qiagen, Hilden, Germany) and Greenstar qPCR Master mix (Bioneer, Daejeon, Korea). Primers were designed by the program Primer3 [24], and the sequences are listed in Table 2. The mRNA level was normalized using GAPDH, a housekeeping gene, and expressed as fold change of the HC group followed by the 2^-ΔΔCt^ method [25]. miR cDNA was synthesized using a miRNA cDNA synthesis kit with poly (A) polymerase tailing (ABM Inc., Richmond, BC, Canada). The cDNA was amplified using specific primers for miR-33, miR-34a, RNU6 (ABM Inc.), and Greenstar qPCR Master mix (Bioneer, Daejeon, Korea). The miRNA level was normalized to RNU6 and expressed as fold change compared with the HC group.

### 2.10. AMP-Activated Protein Kinase (AMPK) Activity

AMPK activity was evaluated using an AMPK Kinase Assay kit (Cyclex, Nagano, Japan) as described previously [26]. Protein concentration was analyzed using a bicinchoninic acid protein assay kit (Thermo Scientific). AMPK activity was normalized to the protein concentration and expressed as fold change relative to the HC group.

### 2.11. Statistical Analysis

All data are expressed as mean ± standard error (SE). The significance of the difference among experimental groups (SD, HC, and PE) was determined by a one-way analysis of variance (ANOVA) followed by Tukey’s multiple comparison tests using SPSS software (version 26; IBM Corporation, Armonk, NY, USA). Significant differences were determined at *p* < 0.05.

## 3. Results

### 3.1. Phenolic Compound Contents of PE

Phenolic compounds in PE were determined by HPLC analysis. Phenolic compounds identified in PE were gallic acid, chlorogenic acid, ferulic acid, and rutin, and the total phenolic content was 5.43 ± 0.20 mg/g (Table 3).

### 3.2. Body Weight, Food Intake, and Serum AST and ALT Activities

At 4 weeks of PE supplementation, body weight, weight gain, and food intake did not differ significantly among the groups (Table 4). HC diet consumption significantly increased liver weight by 33.7% compared with that in the SD group, indicating HC-induced hepatomegaly. However, no significant change was found between the HC and PE groups (Table 4). Serum AST and ALT activities were not significantly different among the groups (Table 4).

### 3.3. Effects of PE on Serum, Liver, and Fecal Lipid Profiles

The serum, liver, and fecal lipids of rats fed SD, HC, and PE diets for 4 weeks are shown in Figure 1 Rats fed an HC diet had higher serum TC and LDL-C concentrations than those fed an SD diet. The PE diet significantly decreased serum TG, TC, and LDL-C concentrations by 31.1%, 29.8%, and 56.9%, respectively, compared with the HC group (*p* < 0.05, Figure 1A). In contrast, HDL-C concentration was higher in the PE group than in the HC group (*p* < 0.05, Figure 1A). The HC diet induced significant increases in hepatic TG and TC levels with larger lipid droplets in the liver tissues, as observed by H&E staining (*p* < 0.05, Figure 1B,C). Hepatic TG and TC levels in the PE group were significantly lower by 22.2% and 10.7%, respectively, than in the HC group (*p* < 0.05, Figure 1C). Fecal levels of TC and TBA were significantly increased by 1.12- and 1.26-fold, respectively, in the PE group compared with the HC group (*p* < 0.05, Figure 1D), but there was no difference in the TG levels.

### 3.4. Effect of PE on Expression of Genes Related to Cholesterol Metabolism

Hepatic mRNA levels involved in cholesterol metabolism were measured by qRT-PCR (Figure 2). Hepatic mRNA levels of LXRα, ABCG5, and ABCG8, which are the key transcriptional regulators of cholesterol efflux in the PE group, were upregulated by 2.35-, 2.21-, and 1.60-fold, respectively, compared with the HC group (*p* < 0.05, Figure 2A). The hepatic mRNA level of CYP7A1, a rate-limiting enzyme in BA synthesis, was upregulated by 2.17-fold in the PE group compared with the HC group (*p* < 0.05, Figure 2B). In contrast, the hepatic mRNA level of FXR, inhibiting excessive BA synthesis, was downregulated by 51.3% in the PE group compared with the HC group (*p* < 0.05, Figure 2B). In the PE group, hepatic mRNA levels of ABCA1, ABCG1, LCAT, apoA-1, SR-B1, and PON1 regulating HDL metabolism were upregulated by 1.95-, 2.10-, 1.51-, 2.04-, 1.93-, and 1.77-fold, respectively, compared with the HC group (*p* < 0.05, Figure 2C).

### 3.5. Effects of PE on AMPK Activity and miR-33/34a Expression

Rats fed an HC diet had 46.3% lower hepatic AMPK activity than rats fed an SD diet. In the PE group, hepatic AMPK activity significantly increased by 1.72-fold compared with the HC group (*p* < 0.05, Figure 3A). Hepatic miR-33 and miR-34a expression levels were significantly higher in the HC group than the SD group. In the PE group, hepatic miR-33 and miR-34a expression levels were significantly lower by 44.9% and 47.6%, respectively, than in the HC group (*p* < 0.05, Figure 3B).

## 4. Discussion

Extrusion elevates the extraction yield by destroying the rigid cell wall [27]. The extrusion procedure maintains a higher total phenolic content of the extrudate, depending on variables such as temperature and moisture [28]. PE contained phenolic compounds such as gallic acid, chlorogenic acid, ferulic acid, and rutin, and the total phenolic content was 5.43 ± 0.20 mg/g. In a previous study, phenolic compounds such as caffeic acid, p-coumaric acid, ferulic acid, quercetin, and kaempferol were identified in the crude extract of PO, and the total amount of phenolic compounds was found to be low (10.32 ± 0.93 mg/kg) [29]. It is assumed that the phenolic content is maintained at a higher level in the extruded extract than in the crude extract of PO. In the present study, we investigated the effect of PE on hepatic cholesterol metabolism in rats fed an HC diet. Furthermore, we confirmed that PE regulates hepatic AMPK through the miR-33/34a pathway. PE supplementation for 4 weeks in rats fed an HC diet improved the serum (TC, HDL-C, and LDL-C), liver (TG and TC), and fecal (TC and TBA) lipid profiles. Similar to our result, administration of aqueous extract of PO in diabetic rats showed a reduction in serum TC and TG levels and increased serum HDL-C levels [30]. Chlorogenic acid and gallic acid improved the lipid profile by lowering the serum levels of TC and TG and increasing the HDL-C levels [31,32]. Our results indicated that PE improved the circulating lipid profile in rats fed an HC diet, suggesting that it is beneficial for improving CVD.

Cholesterol homeostasis is maintained through a balance between cholesterol absorption and excretion, and BA is the end product of cholesterol catabolism. Excretion of BA in the feces reduces excessive hepatic cholesterol [33]. LXRα is an important transcriptional factor that upregulates genes involved in cholesterol catabolism and efflux, such as CYP7A1 and ABCG5/8, in response to excessive cholesterol in the blood [4]. FXR, a negative transcriptional regulator of CYP7A1, enhances the preservation of bile salts [33]. The circulating cholesterol-lowering effects of HDL-C are mainly achieved through the ABCA1 and ABCG1 pathways, which are upregulated by LXRα [33]. ABCA1 facilitates cholesterol efflux through lipid-poor apoA-1, a major apolipoprotein of HDL-C, leading to nascent HDL formation [6]. The nascent HDL matures when cholesterol is esterified under the action of LCAT and receives more cholesterol than can be effluxed by ABCG1 [7]. PON1 is an HDL-C-related enzyme secreted from the liver and regulates the stability of HDL-C as it protects lipoproteins from oxidation [34]. SR-B1 is recognized as an HDL-C receptor and functions to deliver HDL-C into the liver for cholesterol excretion [8]. In the present study, we investigated the expression of genes involved in cholesterol metabolism, including cholesterol efflux, BA synthesis, and HDL-C formation. PE supplementation upregulated the expression of LXRα, ABCG5, ABCG8, and CYP7A1, but downregulated FXR expression. In addition, PE upregulated the gene expression of ABCA1/G1, apoA-1, SR-B1, PON1, and LCAT involved in HDL-C formation, maturation, and uptake. Similar to the beneficial effect of extruded PO extract, the aqueous extract of PO supplementation to rats for 4 weeks enhanced the activity of LCAT and PON1, leading to an increased concentration of serum HDL-C [30]. Treatment of macrophage foam cells with ferulic acid increased the mRNA and protein expression of ABCA1 and ABCG1 [35]. Rutin reduces serum cholesterol levels and upregulates the mRNA level of LXRα in the liver of hamsters [36]. In another study, chlorogenic acid increased TC efflux by upregulating the gene expression of LXRα and CYP7A1 in human hepatoma HepG2 cells [37]. Our results suggest that PE positively affects cholesterol efflux, BA synthesis, and HDL-C formation, which may be beneficial for treating hypercholesterolemia.

AMPK, an enzyme that acts as a sensor in maintaining energy homeostasis, is activated when the cellular energy responding to low ATP levels is reduced because of the nutrient status [38]. In particular, AMPK inhibition accelerates the development and progression of atherosclerosis [39]. AMPK activation significantly upregulated the expression of ABCA1/G1 in macrophages and SR-B1 and LCAT in the liver [40]. Phenolic compounds such as ferulic acid, chlorogenic acid, and rutin increased AMPK activity [41,42,43]. We found that AMPK activity was increased by PE supplementation. Thus, it appears that the therapeutic efficacy of PE is due to the activation of AMPK by its phenolic compounds.

miRs are small single-stranded RNAs consisting of 21–25 nucleotides and are regulators of gene expression in eukaryotes [44]. miR-33 and miR-34a are key regulators of HDL-C biogenesis thanks to their inhibitory effect on gene expression of regulators of HDL-C metabolism such as LXRα and ABCA1/G1 [23,45]. Moreover, it was reported that miR-34a could directly inhibit the phosphorylation of AMPK [46]. In this study, PE significantly reduced the hepatic miR-33 and miR-34a levels. Gallic acid has been shown to have an inhibitory effect on miR-34a expression, and ferulic acid is also known to downregulate both miR-33 and miR-34a levels [47,48]. In this study, we found for the first time that PE regulates hepatic miR-33 and miR-34a expression in rats fed an HC diet. Therefore, it can be suggested that PE partially improves the hepatic lipid profile via miR-33 and miR-34a regulation.

## 5. Conclusions

In conclusion, our results suggest that PE improves the serum, liver, and fecal lipid profiles and positively alters the expression of genes involved in hepatic cholesterol metabolism in rats fed an HC diet. Moreover, we report for the first time that the lipid-improving effect of PE may be partially involved with increased AMPK activity and miR-33/34a inhibition in the liver (Figure 4). Therefore, we suggest that PE may be useful as a potential functional food for improving CVD. As this study limited the scope of application to animals, it is considered that future studies on the clinical application and safety of PE are necessary to maximise the use of PE as a source of bioactive components.

## Figures and Tables

**Figure 1 nutrients-14-03330-f001:**
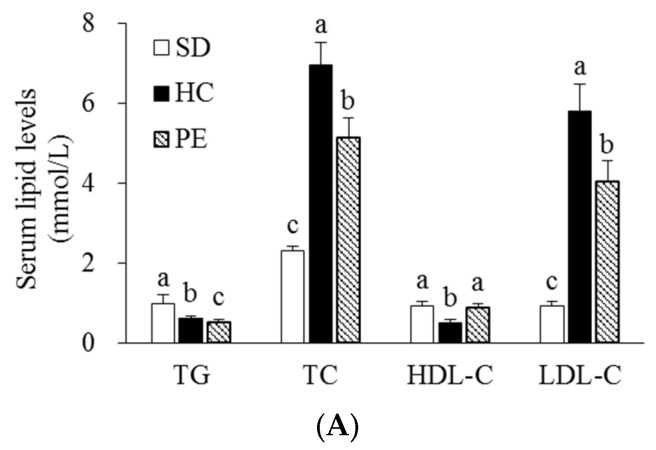
Effects of PE on serum (**A**), hepatic (**C**), and fecal lipid profiles (**D**). (**B**) Representative H&E stained liver tissue (scale bars = 100 μm; magnification of 400×). Data are expressed as mean ± SE (*n* = 8). Bars with different letters (a, b, c) are statistically different at *p* < 0.05 by Tukey’s test. SD, standard diet; HC, high-cholesterol; PE, HC + 0.8% extruded *Portulaca oleracea* L. extract; HDL-C, high-density lipoprotein-cholesterol; LDL-C, low-density lipoprotein-cholesterol; TBA, total bile acid; TC, total cholesterol; TG, triglyceride; CV, central vein; L, lipid deposition.

**Figure 2 nutrients-14-03330-f002:**
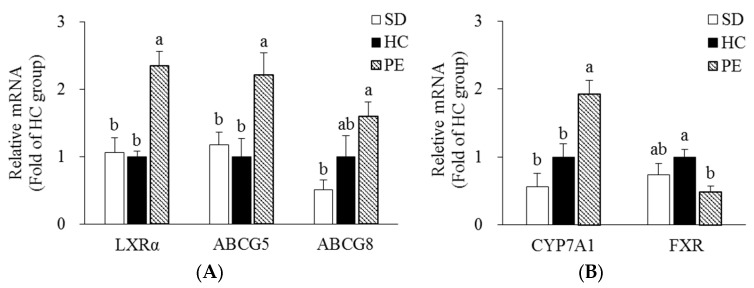
Effects of PE on mRNA levels of genes related to cholesterol metabolism in the liver. (**A**) The mRNA levels of LXR-α, ABCG5, and ABCG8. (**B**) The mRNA levels of CYP7A1 and FXR. (**C**) The mRNA levels of ABCA1, ABCG1, LCAT, apoA-1, SR-B1, and PON1. Data are expressed as fold change compared with the HC group and expressed as mean ± SE (*n* = 8). Bars with different letters (a, b) are statistically different at *p* < 0.05 by Tukey’s test. SD, standard diet; HC, high-cholesterol diet; PE, HC + 0.8% extruded *Portulaca oleracea* L. extract. ABCA1, ATP-binding cassette subfamily A1; ABCG1, ATP-binding cassette subfamily G1; ABCG5, ATP-binding cassette subfamily G member 5; ABCG8, ATP-binding cassette subfamily G member 8; apoA-1, apolipoprotein A-l; CYP7A1, cholesterol 7 alpha hydroxylase; FXR, farnesoid X receptor; GAPDH, glyceraldehyde 3-phosphate dehydrogenase; LCAT, lecithin-cholesterol acyltransferase; LXR, liver X receptor alpha; PON1, paraoxonase 1; SR-B1, scavenger receptor class B type 1.

**Figure 3 nutrients-14-03330-f003:**
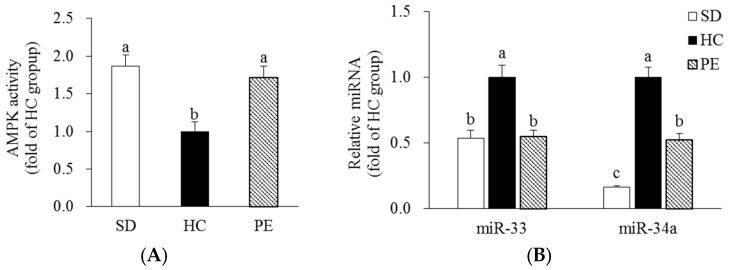
Effect of PE on AMPK activity (**A**) and miR-33/34a expression (**B**) in the liver. Data are expressed as mean ± SE (*n* = 8). Bars with different letters (a, b, c) are statistically different at *p* < 0.05 by Tukey’s test. SD, standard diet; HC, high-cholesterol diet; PE, HC + 0.8% extruded *Portulaca oleracea* L. extract; AMPK, AMP-activated protein kinase; miR, microRNA.

**Figure 4 nutrients-14-03330-f004:**
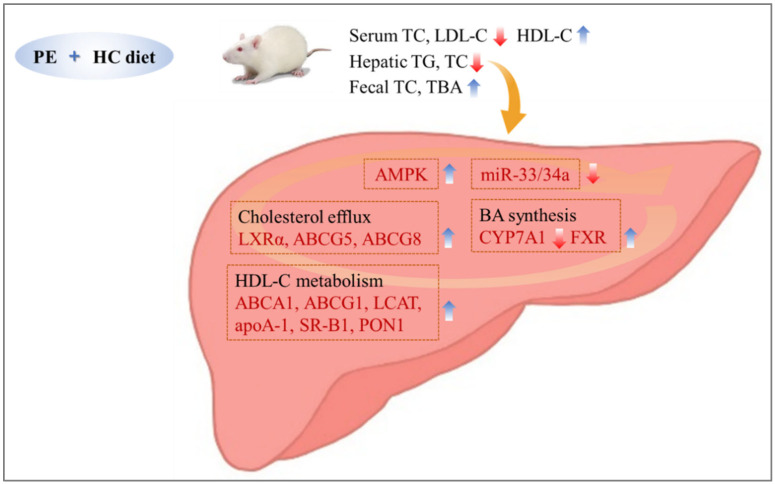
Schematic diagram showing potential regulatory mechanisms in the liver for the lipid-improving effects of PE in rats fed a high-cholesterol diet. HC, high-cholesterol; PE, 0.8% extruded *Portulaca oleracea* L. extract; ABCA1, ATP-binding cassette subfamily A1; ABCG1, ATP-binding cassette subfamily G1; ABCG5, ATP-binding cassette subfamily G member 5; ABCG8, ATP-binding cassette subfamily G member 8; AMPK, AMP-activated protein kinase; apoA-1, apolipoprotein A-l; CYP7A1, cholesterol 7 alpha hydroxylase; FXR, farnesoid X receptor; HDL-C, high-density lipoprotein-cholesterol; LCAT, lecithin-cholesterol acyltransferase; LDL-C, low-density lipoprotein-cholesterol; LXRα, liver X receptor alpha; miR, microRNA; PON1, paraoxonase 1; SR-B1, scavenger receptor class B type 1; TBA, total bile acid; TC, total cholesterol; TG, triglyceride.

**Table 1 nutrients-14-03330-t001:** Composition of experimental diets (g/kg).

Component	SD	HC	PE
Corn starch	150	150	150
Casein	200	200	200
Sucrose	500	485	477
Corn oil	50	50	50
Cellulose	50	50	50
Mineral mix (AIN-76)	35	35	35
Vitamin mix (AIN-76)	10	10	10
DL-Methionine	3	3	3
Choline bitartrate	2	2	2
Cholesterol	-	10	10
Cholic acid	-	5	5
PO extract	-	-	8
Total	1000	1000	1000

SD, standard diet; HC, high cholesterol; PE, HC + 0.8% PO extract. This experimental diet was formulated based on a modification of the committee of the American Institute of Nutrition (AIN)-76 diet composition [20].

**Table 2 nutrients-14-03330-t002:** Primer sequences used for RT-qPCR.

Name	Genebank No.	Primer Sequence (5′–3′)
ABCA1	NM_178095.3	F: GCT ACA CTG GTC GTT ATC AT R: GAC CAC CCA TAT AGC AAG AG
ABCG1	NM_053502.2	F: GCT CTG TGG AGG TAG TTA ATG R: CTC CTT CCA GAC TTC CTT TC
ABCG5	NM_053754	F: ATG AGT GAG CTG CCC TTT CT R: CGC TGA AGG ACA CAT TCA GG
ABCG8	NM_130414.2	F: CAC CCT AGA CTC TAA CTC CA R: GGA GCA CTG GAT AGT ATT GG
apoA-1	NM_012738	F: GTG GGT TCA ACT GTT GGT CG R: GGG CTG CAT CTT CTG TTT CA
CYP7A1	NM_012942.2	F: GCC TTC CTA TTC ACT TGT TC R: GTG GAG AGC GTG TCA TTG
FXR	NM_021745	F: TTC ACT GTC TGA TCC GCA TG R: CGC CGT GTA CAA GTG TAA GA
GAPDH	NM_017008.4	F: GAT GAC ATC AAG AAG GTG GT R: GCA TCA AAG GTG GAA GAA TG
LCAT	NM_017024.2	F: TAA CAA TGG GTA TGT GCG GG R: GCC AAG GCT GTG TCC AAT AA
LXRα	NM_031627	F: GAC TTC GAG TCA CGC CTT GG R: GTC CTC CCT GCT CAG CTG TA
PON1	BC_091403.1	F: GT GGT AAT CCA CCC AGA CTC R: AA GCT CTC AGG TCC AAC AGC
SR-B1	NM_031541	F: GG CAA ATT TGG CCT GTT CGT R: GC CGT TCC ACT TAT CCA CCA

ABCA1, ATP binding cassette subfamily A1; ABCG1, ATP binding cassette subfamily G1; ABCG5, ATP binding cassette subfamily G member 5; ABCG8, ATP binding cassette subfamily G member 8; apoA-1, apolipoprotein A-l; CYP7A1, cholesterol 7 alpha hydroxylase; FXR, farnesoid X receptor; GAPDH, glyceraldehyde 3-phosphate dehydrogenase; LCAT, lecithin-cholesterol acyltransferase; LXR, liver X receptor alpha; PON1, paraoxonase 1; SR-B1, scavenger receptor class B type 1.

**Table 3 nutrients-14-03330-t003:** Phenolic compounds identified in PE.

Compound	Content (mg/g)
Gallic acid	1.13 ± 0.02
Chlorogenic acid	2.52 ± 0.16
Ferulic acid	0.41 ± 0.01
Rutin	1.37 ± 0.02

Data are expressed as mean ± SE of three replicates by *t*-test. PE, extruded *Portulaca oleracea* L. extract.

**Table 4 nutrients-14-03330-t004:** Effect of PE on physiological variables.

Variables	SD	HC	PE
Initial body weight (g)	191.25 ± 1.25	191.65 ± 2.00	190.64 ± 1.68
Final body weight (g)	404.40 ± 6.95	402.25 ± 2.62	403.19 ± 6.12
Body weight gain (g/4 weeks)	213.14 ± 6.90	210.59 ± 3.24	212.55 ± 5.60
Food intake (g/day)	24.93 ± 0.52	24.60 ± 0.82	23.72 ± 0.45
Liver weight (g/100 g body weight)	3.15 ± 0.06	4.75 ± 0.17	4.63 ± 0.14
Serum ALT (IU/L)	5.82 ± 0.50	7.03 ± 0.86	6.37 ± 0.58
Serum AST (IU/L)	48.02 ± 1.86	45.69 ± 3.07	44.04 ± 2.82

Data are expressed as mean ± SE (*n* = 8). Significant differences were analyzed by Tukey’s test. SD, standard diet; HC, high cholesterol; PE, HC + 0.8% extruded *Portulaca oleracea* L. extract; ALT, alanine transaminase; AST, aspartate aminotransferase.

## Data Availability

The data supporting the findings of this study are available upon request from the corresponding authors. The data are not publicly available for privacy reasons.

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
