# Peer review of "Portulaca oleracea L. Extract Regulates Hepatic Cholesterol Metabolism via the AMPK/MicroRNA-33/34a Pathway in Rats Fed a High-Cholesterol Diet"

_nutrients, 2022, doi:10.3390/nu14163330_

Round 1

Reviewer 1 Report

The manuscript investigated and interesting and actual topic of research. I have some suggestions to improveits quality

line 136: have the primers being designed by authors? 

line 138: how did the author check housekeeeing gene stability and suitability in theor experiemntal conditions?

line 142: why did the author normalized on RNU6?

Figure 1: i suggest to move to supplementary informations as it provides similar info of table 3.

line 163 and line 250. the author reported the total phenolic content of 5.43. can you discuss the quality of phenols in PE? is it similar to other phenolic sources?

Line 308: other studies are requested to confirm and corroborate PE activity before their inclusion in feed/ food.

General consideration. colesterol lowering activity can be due to the presence of other bioactive components in PE? or only to Polyphenols identified by the authors? 

also, even if PE has a cosmopolitan distribution etc, I suggest the authors o mention the limits and potential measures to maximise the use of PE as source of bioactive components.  A discuss on the latter points is expected.

Author Response

Manuscript Number: nutrients-1837054

Title: Portulaca Oleracea L. Extract Regulates Hepatic Cholesterol Metabolism via AMPK/MicroRNA-33/34a Pathway in Rats Fed a High-Cholesterol Diet

Responses to Reviewer's comments (Reviewer 1)

The manuscript investigated and interesting and actual topic of research. I have some suggestions to improveits quality

We thank the reviewer for careful reading and description about our manuscript with the valuable comments. We worked to the best of our abilities to revise the issues reviewer point out.

line 136: have the primers being designed by authors?

Response: Thank you for your comments, we revised as follows on Page 4.

“Primers were designed by the program Primer3 [23] and the sequences are listed in Table 2.”

Reference

[23] Rozen S, Skaletsky H. Primer3 on the WWW for general users and for biologist programmers. Methods Mol Biol. 2000;132:365-86. doi: 10.1385/1-59259-192-2:365.

line 138: how did the author check housekeeping gene stability and suitability in theor experiemntal conditions?

Response: We really appreciate reviewer’s valuable comment. When performing quantitative real-time PCR (qRT-PCR), it is the most importance to select a stable reference gene to normalize target mRNA levels. In the present study, we optimized the conditions for not only target genes but also housekeeping gene (GAPDH) by PCR efficiency including g slope, R2, and melt curve with a serial dilution of pooled samples. In addition, the influence on housekeeping gene level in the liver of the experimental groups was analyzed by consistent Ct values among 3 groups and electrophoresis.

line 142: why did the author normalized on RNU6?

Response: Thank you for your helpful comments. RNU6, a U6 small nuclear RNA, is commonly used as an endogenous control when studying miRNAs [1-4]. In addition, in the liver tissue of this study, RNU6 was used as a reference gene because it was expressed while maintaining a stable Ct value.

Reference:

[1] Singh A, Kant R, Nandi S, Husain N, Naithani M, Mirza AA, Saluja TS, Srivastava K, Prakash V, Singh SK. Detection of differential expression of miRNAs in computerized tomography-guided lung biopsy. J Cancer Res Ther. 2022;18(1):231-239. doi: 10.4103/jcrt.jcrt_749_21.

[2] Wang J, Zhang J, Ding X, Wang Y, Li Z, Zhao W, Jia J, Zhou J, Ge J. Differential microRNA expression profiles and bioinformatics analysis between young and aging spontaneously hypertensive rats. Int J Mol Med. 2018;41(3):1584-1594. doi: 10.3892/ijmm.2018.3370.

[3] Akbari G, Mard SA, Dianat M, Mansouri E. The Hepatoprotective and MicroRNAs Downregulatory Effects of Crocin Following Hepatic Ischemia-Reperfusion Injury in Rats. Oxid Med Cell Longev. 2017;2017:1702967. doi: 10.1155/2017/1702967.

[4] Sangiao-Alvarellos S, Pena-Bello L, Manfredi-Lozano M, Tena-Sempere M, Cordido F. Perturbation of hypothalamic microRNA expression patterns in male rats after metabolic distress: impact of obesity and conditions of negative energy balance. Endocrinology. 2014;155(5):1838-50. doi: 10.1210/en.2013-1770.

Figure 1: i suggest to move to supplementary informations as it provides similar info of table 3.

Response: As you suggested, Figure 1 has been moved to supplementary information.

Page 3:

“The HPLC chromatogram of the phenolic compounds is presented in Supplementary Figure S1.”

line 163 and line 250. the author reported the total phenolic content of 5.43. can you discuss the quality of phenols in PE? is it similar to other phenolic sources?

Response: We sincerely appreciate the reviewer's comments. As you suggested, we discussed as follows on Page 9.

“PE contained phenolic compounds such as gallic acid, chlorogenic acid, ferulic acid, and rutin, and the total phenolic content was 5.43 ± 0.20 mg/g. In a previous study, phenolic compounds such as caffeic acid, p-coumaric acid, ferulic acid, quercetin, and kaempferol were identified in the crude extract of PO, and the total amount of phenolic compounds was found to be low (10.32 ± 0.93 mg/kg) [28]. It is assumed that the phenolic content is maintained higher in the extruded extract than in the crude extract of PO.

Line 308: other studies are requested to confirm and corroborate PE activity before their inclusion in feed/ food.

Response: Thank you for your comments. We added in the Discussion section as follows:

Page 10:

“Therefore, we suggest that PE may be useful as a potential functional food for improving CVD. Since this study limited the scope of application to animals, it is considered that future studies on the clinical application and safety of PE are necessary to maximise the use of PE as source of bioactive components.

General consideration.

cholesterol lowering activity can be due to the presence of other bioactive components in PE? or only to Polyphenols identified by the authors?

Response: Thank you for your comments. We further described other bioactive components of PE in the Introduction section.

Page 2:

“PO has been used as a traditional medicine for centuries in many countries [14]. It has many bioactive components, including polyphenols, flavonoids, alkaloids, vitamins, minerals, and omega-3 fatty acid [15-18], and it is the most abundant in the alpha-linolenic acid among vegetables [18].”

also, even if PE has a cosmopolitan distribution etc, I suggest the authors mention the limits and potential measures to maximise the use of PE as source of bioactive components. A discuss on the latter points is expected.

Response: We truly appreciate the reviewer's suggestion. We have additionally mentioned in the Discussion section as follows:

Page 10:

“Since this study limited the scope of application to animals, it is considered that future studies on the clinical application and safety of PE are necessary to maximise the use of PE as source of bioactive components.”

Reviewer 2 Report

There are some major and minor comments that should be adressed.

The term "polyphenol" should be restricted to structures bearing at least two phenolic moieties, despite of the number of hydroxil groups (Quideau et al., 2021 DOI: 10.1002/anie.201000044). Considering that, gallic acid and ferulic acid are not polyphenol. Please, make adjust in all text.

"Methods section"

1. describe the parameters used to identify the phenolic compounds in PE by HPLC.

2. describe how the quantification of phenolics was performed.

3. Describe in detail the method of euthanasia and sample collection (blood and liver).

"Results section"

1. Provide write footnotes of table and figures for statistical tests, etc

2. Line 290- rutin is not a phenolic acid. It´s belong to flavonoid class.

"Discussion section"

1. Explain the strenghts and weakness of the study.

Author Response

Manuscript Number: nutrients-1837054

Title: Portulaca Oleracea L. Extract Regulates Hepatic Cholesterol Metabolism via AMPK/MicroRNA-33/34a Pathway in Rats Fed a High-Cholesterol Diet

Responses to Reviewer's comments (Reviewer 2)

There are some major and minor comments that should be adressed.

We thank the reviewer for careful reading and description about our manuscript with the valuable comments. We worked to the best of our abilities to revise the issues reviewer point out.

The term "polyphenol" should be restricted to structures bearing at least two phenolic moieties, despite of the number of hydroxil groups (Quideau et al., 2021 DOI: 10.1002/anie.201000044). Considering that, gallic acid and ferulic acid are not polyphenol. Please, make adjust in all text.

Response: We truly appreciate the reviewer's suggestion. We revised in all text as follows:

Page 3:

“Phenolic compounds in PE were identified and quantified using the Agilent 1100 HPLC System (CA, USA).”

Page 6:

“3.1. Phenolic Compound Contents of PE

Phenolic compounds in PE were determined by HPLC analysis. Phenolic compounds identified in PE were gallic acid, chlorogenic acid, ferulic acid, and rutin, and the total phenolic content was 5.43 ± 0.20 mg/g (Table 3).

“Table 3. Phenolic compounds identified in PE.”

Page 9:

“PE contained phenolic compounds such as gallic acid, chlorogenic acid, ferulic acid, and rutin, and the total phenolic content was 5.43 ± 0.20 mg/g.”

"Methods section"

  1. describe the parameters used to identify the phenolic compounds in PE by HPLC.

Response: Thank you for your comments. As you suggested, we added in the Methods section as follows:

Page 3:

“The regression equation was obtained from the analysis results of phenolic compound standards (Supplementary Table S1). The phenolic content of PE was obtained by substituting y as area and x as quantity. As a result of HPLC analysis of PE, the peaks of gallic acid at 9.402 min, chlorogenic acid at 31.560 min, ferulic acid at 48.804 min, and rutin at 55.649 min were confirmed.”

Supplementary Table 1. Chromatographic parameters of phenolic compound standards analyzed by HPLC

Compound

Regression equation

r2

Gallic acid

y=25.954x-220.916

0.999

Chlorogenic acid

y=16.830x-127.714

0.999

Ferulic acid

y=66.322x-143.433

0.999

Rutin

y=10.276x-7.209

0.999

  1. describe how the quantification of phenolics was performed.

Response: Thank you for your comments. As you suggested, we added in the Methods section as follows:

Page 2, 3:

“ 2.2. Sample Preparation for HPLC Analysis

As phenolic compound standards, gallic acid, chlorogenic acid, ferulic acid, and rutin were purchased from Sigma-Aldrich (St. Louis, MO, USA) were used for the determination of phenolic content in PE. Precisely weighed 0.01 g of the standard in a 50 mL conical tube, added 10 mL of methanol to yield a standard concentration 1000 ppm, and dissolved throughly in a vortex mixer for 5 minutes. It was then diluted to appropriate concentrations ranged 62.5, 125, 250 and 500 ppm. Finally, the di-lutes were filtered through a 0.20 µm polytetrafluoroethylene syringe filter. In order to determine the phenolic content of PE, weighed accurately 0.1 g of PE powder in a 50 mL conical tube, and added 10 mL of distilled water. And then dissolved in a voltex mixer for 5 minutes. Finally, the solution was filtered through a 0.20 µm polytetrafluoethylene syringe filter. Phenolic compounds in samples were quantified based on calibration with standards.”

  1. Describe in detail the method of euthanasia and sample collection (blood and liver).

Response: As you suggested, we added the method of euthanasia and sample collection.

Page 3:

“At the end of the 4-week of PE supplementation, all rats were anesthetized with a mixture of Zoletil 50 (Virbac Laboratories, Carros, France) and Rompun (Bayer Korea, Seoul Korea) after 12 h overnight fasting. Liver tissue was excised through an abdominal midline incision, weighed immediately, frozen in liquid nitrogen, and stored at −70°C. Blood samples were collected by cardiac puncture, separated by centrifugation at 3,000rpm for 20 min, and stored at −70°C.”

"Results section"

  1. Provide write footnotes of table and figures for statistical tests, etc

Response: As you suggested, we have added statistical test methods (T-test or Tukey’s test) to the footnotes of figures and tables.

  1. Line 290- rutin is not a phenolic acid. It´s belong to flavonoid class.

Response: Thank you for your comments, we revised as follows on Page 10.

“Phenolic compounds such as ferulic acid, chlorogenic acid, and rutin increased AMPK activity [38-40].”

"Discussion section"

  1. Explain the strenghts and weakness of the study.

Response: Thank you for your comments. We added in the Discussion section as follows:

Page 10:

“Moreover, we report for the first time that the lipid-improving effect of PE may be partially involved with increased AMPK activity and miR-33/34a inhibition in the liver (Figure 4). Therefore, we suggest that PE may be useful as a potential functional food for improving CVD. Since this study limited the scope of application to animals, it is considered that future studies on the clinical application and safety of PE are necessary to maximise the use of PE as source of bioactive components.”
